# A Bayesian Approach to Describe and Simulate the pH Evolution of Fresh Meat Products Depending on the Preservation Conditions

**DOI:** 10.3390/foods11081114

**Published:** 2022-04-13

**Authors:** Ngoc-Du Martin Luong, Louis Coroller, Monique Zagorec, Nicolas Moriceau, Valérie Anthoine, Sandrine Guillou, Jeanne-Marie Membré

**Affiliations:** 1Oniris, INRAE, SECALIM, 44200 Nantes, France; ngoc-du.luong@oniris-nantes.fr (N.-D.M.L.); monique.zagorec@oniris-nantes.fr (M.Z.); nicolas.moriceau@oniris-nantes.fr (N.M.); valerie.anthoine@oniris-nantes.fr (V.A.); sandrine.guillou@inrae.fr (S.G.); 2Univ Brest, INRAE, Laboratoire Universitaire de Biodiversité et Écologie Microbienne, UMT ACTIA Alter’iX 19.03, 29000 Quimper, France; louis.coroller@univ-brest.fr

**Keywords:** poultry sausage, potassium lactate, modified atmosphere packaging, food modelling, nonlinear model, Bayesian inference

## Abstract

Measuring the pH of meat products during storage represents an efficient way to monitor microbial spoilage, since pH is often linked to the growth of several spoilage-associated microorganisms under different conditions. The present work aimed to develop a modelling approach to describe and simulate the pH evolution of fresh meat products, depending on the preservation conditions. The measurement of pH on fresh poultry sausages, made with several lactate formulations and packed under three modified atmospheres (MAP), from several industrial production batches, was used as case-study. A hierarchical Bayesian approach was developed to better adjust kinetic models while handling a low number of measurement points. The pH changes were described as a two-phase evolution, with a first decreasing phase followed by a stabilisation phase. This stabilisation likely took place around the 13th day of storage, under all the considered lactate and MAP conditions. The effects of lactate and MAP on pH previously observed were confirmed herein: (i) lactate addition notably slowed down acidification, regardless of the packaging, whereas (ii) the 50%CO_2_-50%N_2_ MAP accelerated the acidification phase. The Bayesian modelling workflow—and the script—could be used for further model adaptation for the pH of other food products and/or other preservation strategies.

## 1. Introduction

Spoilage occurrence in fresh meat products characterized by changes in their sensory quality during storage is usually due to the development of microorganisms contaminating the products or to chemical deterioration, such as oxidation [1]. Studying spoilage during storage is generally done by monitoring several physicochemical or microbiological spoilage responses [2,3]. Among these responses, pH changes in meat products have been widely measured, since pH is known as an intrinsic factor that can both influence and change, depending on the growth of several spoilage-associated microorganisms [1,4,5]. For instance, the development of Lactic Acid Bacteria (LAB) is intimately associated with pH changes, since these bacteria can behave differently under different pH conditions and also produce lactic acid by their fermentative activities, leading to the acidification of meat products [6,7]. Analyzing acidification during storage by describing pH changes may represent a simple first step in studying meat spoilage, since pH can be measured easily and accurately on meat in the laboratory and also reflect the development of several microorganisms frequently considered as spoilage indicators. Besides, several strategies can be used by food manufacturers to limit the development of these microorganisms. We can cite, for example, adding food preservatives, including lactate, or packaging products under modified atmosphere (MAP), with mixtures of O_2_ and CO_2_ gases [8,9,10,11,12,13]. In the field of predictive microbiology, the effects of these strategies are generally described by modelling the behavior of the target microorganisms [14,15,16,17]. However, the application of such approaches for the spoilage of fresh meat products is not straightforward when spoilers have not been identified beforehand in the whole food microbiota [16]. Besides, modelling different responses associated with spoilage can also be difficult due to sparse experimental data.

In this context, Bayesian modelling approaches for describing pH changes can be an interesting way to monitor spoilage. Firstly, Bayesian methods can help to fill data gaps with prior information, possibly provided by experts or other preliminary works [18,19]. Secondly, such methods can be useful for the analyses of food products to decipher potential sources of variability and/or uncertainty, since the conception of Bayesian models can include these sources as parameters [20]. Thirdly, the results of parameter estimations, obtained as posteriori value distributions, could be useful for further model validation using external data, as well as for numerical simulations of pH under other conditions.

This study aimed to develop a Bayesian modelling approach to estimate parameters of acidification kinetics, with a limited number of time points in the monitored pH curves. Bayesian modelling has already been applied to meat matrices [21,22]. Hereby, it was applied to fresh sausages, industrially made using different potassium lactate formulations and modified atmospheres for packaging. The pH factor was chosen as spoilage response because (i) pH is closely linked to spoilage-associated microorganisms, (ii) pH is measurable with good precision in the laboratory and (iii) pH prior information required for the Bayesian approach can be obtained by experts or from previous works in the literature.

## 2. Materials and Methods

### 2.1. Products and pH Measurement

The experimental data used in this study were extracted from a data paper monitoring several spoilage responses of fresh turkey sausages with different lactate formulations and conditioned under modified atmospheres during chilled storage (the data paper is available online [23]). Only data associated with the monitoring of pH were used herein. The meat of the sausages was from turkey. Pork fat was added to a final content of 11% in sausages. Potassium lactate (2.0% *w/w*, corresponding to full normal dose) was used as additive. Furthermore, sausages were battered with a spice mix at a concentration of ∼2.5%; it contained sodium salt ∼40%, dextrose ∼10%, spices ∼15%, aromas ∼22%. Sausages were produced in ten independent batches and under several “process conditions”. These conditions corresponded to three initial concentrations of potassium lactate in sausage formulation combined with three modified atmosphere packaging processes. In practice, for each production batch, sausages were first made from the same batch of turkey meat in sufficient quantity for all conditions. The sausages were provided by two French meat processing companies, and the total quantity of sausages produced within the research project was around 360 kg. Indeed, since within the project microbial, physico-chemical and sensorial analyses on sausages were performed, four different trays with five sausages per tray (equivalent of 5 × 50 g of sausages per tray) were required. Nine conditions were considered in the present study: for simplification purpose, in the present article, we denoted “Full dose”, “Half dose” and “Zero lactate” for the three lactate formulations (2%, 1% and 0% *w/w* in sausage samples, respectively); “Air”, “MAP_1_” and “MAP_2_” for the three packaging conditions (Air packaging, MAP_1_: 70%O_2_-30%CO_2_ and MAP_2_: 50%CO_2_-50%N_2_, respectively). The level of lactate and type of modified atmosphere were chosen as function of current practice in industry in France. For each condition, analyses were done at four time points (day 2, 8, 15, 22 after production). Therefore, the required minimum quantity of sausages was 50 g (sausage) × 5 (sausages/tray) × 4 (trays) × 3 (atmospheres) × 3 (lactate formulation) × 4 (sampling time) = 36 kg per batch. All the above quantities were repeated in ten independent industrial batches which makes finally 360 kg of sausages.

The pH measurement was done in each condition using the tray destined for microbial analyses. All pH measurements were done in triplicate (technical replicates) for each of four sampling time points. The choice of these points was inspired by the French Norm NF V01-003 used to establish the shelf life of chilled perishable and highly perishable food [24]. It consists of storing samples at 4 °C for one-third of the shelf life and then, at 8 °C during the remaining two-thirds of the storage duration. The use-by-date (UBD) established by the producers for the sausages containing the normal dose of lactate and conditioned under modified atmosphere, was 15 days for turkey sausages. Therefore, sausages were first stored at 4 °C for 5 days and then placed at 8 °C during the remaining storage duration. To be able to observe spoilage occurrence, the storage duration was extended to one UBD and half, i.e., 22 days, hence the four chosen sampling time points at 2, 8, 15 and 22 days after production, respectively. The data associated with pH are plotted in Figure 1.

### 2.2. Formalisation of the Model

First, it should be noted that not all the pH curves were independent because the different formulations and packaging steps were carried out on the samples from the same batch after production. Second, samples under each process condition were monitored at different times. Consequently, the model was built with successive levels: (i) production batch, (ii) process condition and (iii) time of sampling (Figure 2).

#### 2.2.1. Deterministic Part

We denote by $N_{R},N_{P},N_{T}$ the total number of production batches, the total number of atmosphere conditions and the total number of sampling time points, respectively $\left(N_{R}=10,\;N_{P}=3,\;N_{T}=4\right)$. Within a production batch $r\;\left(r=1,\ldots ,N_{R}\right)$, the initial pH value, denoted $pH_{0}$, was assumed to be fixed for a batch for all process conditions since sausage samples were formulated and packed from each batch.

Starting from the same initial value $pH_{0}$, the acidification processes under different conditions were described by a two-phase model, based on the patterns observed in the data. For a given ‘process’ (lactate and packaging) condition k (k=1,…), the deterministic part of the model assumed that pH values linearly decreased with an acidification rate $\beta_{k}$ (expressed as unit of pH decreasing per day), supposed to vary from one process condition to another, before reaching a stabilisation phase from the time point θ (expressed in days).

The acidification rate $\beta_{p}$, supposed to vary depending on the initial lactate content and the atmosphere for packaging p, was modelled using log-linear-based models as follows:(1)$$\begin{array}{c} \ln\left(\beta_{P}\right)=\lambda .\;Lactate^{n}+\delta_{P}, \end{array}$$
where Lactate is the initial lactate content (expressed in % *w/w*), $\beta_{P}$ (unit of pH decreasing per day, $\beta_{P}>0$) is the acidification rate of the formulated sausages under the atmosphere p (p=Air, MAP1,MAP2), λ is a slope parameter describing effect of lactate on the acidification rate, $\delta_{P}$ represents the additive effect of the atmosphere p on the acidification rate, and n is a scale parameter (log-linear model in case of n=1).

#### 2.2.2. Stochastic Part

The stochastic part of the model aimed to describe random variations due to variabilities and uncertainties by assuming probability distributions for model parameters. An initial statistical analysis of the dataset revealed a significant batch effect: pH changes were different from one batch to another [25]. This batch effect was then taken into account in the present paper. The initial value $pH_{0}$ after production was assumed to vary between production batches and to follow then a normal distribution:(2)$$\begin{array}{c} pH_{0}∼N\left(\mu_{pH_{0}},\sigma_{pH_{0}}\right), \end{array}$$
where $\mu_{pH_{0}}$ and $\sigma_{pH_{0}}$ corresponded to the mean and standard deviation of the pH observed in fresh sausages at the day of production.

For a given lactate-atmosphere condition k, the pH of sausages at the $time_{i}\left(i=1,\ldots ,\;N_{T}\right)$, denoted $Y_{i\left(k\right)},$ was modelled as follows:(3)$$\begin{array}{c} Y_{i\left(k\right)}∼N\left(m_{i\left(k\right)};\sigma_{pH}\right) \end{array}$$
where $\sigma_{pH}$ represents the random variations in the measurement at each time point and $m_{i\left(k\right)}$ the expected pH value under given process condition and time:(4)$$\begin{array}{c} m_{i\left(k\right)}=\{\begin{array}{c} -\beta_{k}.time_{i\left(k\right)}+pH_{0},\;\;\forall \;time_{i}<\theta \;(\mathrm{acidification}\;\mathrm{phase})\\ -\beta_{k}.\;\theta +pH_{0},\;\forall time_{i}\geq \theta \;\left(\mathrm{stabilisation}\;\mathrm{phase}\right) \end{array} \end{array}$$

With $\beta_{k}$ the acidification rate of samples under the condition k, $pH_{0}$ the initial pH value as described in Equation (2), θ the stabilisation time (in days). The description of all model parameters is gathered in Table 1.

### 2.3. Parameters Inference

Prior distributions were specified for each model parameter and are given in Table 1. First, the prior for initial mean value of pH $\mu_{pH_{0}}$ was defined from data extracted from previous work of Lerasle, et al. [26]. The authors provided a mean and standard deviation of the pH in fresh poultry sausages at 5.84 and 0.11, respectively. The prior of $\mu_{pH_{0}}$ was then described by a normal distribution N(5.84, 0.11). The standard deviation parameters $\sigma_{pH_{0}}$ and $\sigma_{pH}$ were described by half-normal distributions half-N(0,0.1) (positive value ranges) for simplification purposes [19]. The effects of lactate and atmospheres were described using uniform distributions.

The main difficulty in the estimation of θ (stabilisation time) and n (scale parameter characterising effects of lactate, Equation (2) resides mainly in the adjustment of kinetic models in a low number of experimental points (3 formulations and 4 measurements across 22 days). Therefore, a comparative strategy was carried out to better estimate these parameters. For this, by principle of parsimony, we considered four (sub)-models where θ and n were alternatively fixed or to be estimated (see Table 1). When applicable, θ was arbitrarily set at 15 days, corresponding to the use-by-date (UBD) established by the producers for sausages; otherwise, θ was assumed to follow a uniform distribution with an interval of ±7 days around the above prior UBD, θ∼U(8;22). Finally, where appropriate, n was set at n=1 (corresponding to a simple log-linear first order to describe the acidification rates); otherwise, if estimated, the prior of n was described by a uniform distribution.

Bayesian inference was performed using the JAGS software linked to the rjags R package [27,28]. Three independent Markov Chain Monte Carlo (MCMC) chains starting at different given values were run in parallel. After an initial burn-in period of 5000 iterations, for each chain, the Bayesian algorithm simulated 60,000 iterations and sampled from these runs with a thinning interval of six iterations in order to reduce autocorrelations between consecutive iterations. The final sample size per MCMC chain was then 10,000. The total number of iterations for the above simulations and thinning was chosen using a method proposed by Raftery and Lewis [29] who calculated a minimum effective sample size required from a short preliminary test run. The convergence of the estimation process was checked by visually diagnosing the similarity between the ranges of output values of the three chains and by the convergence criteria proposed by Gelman and Rubin [30] and Geweke [31]. Samples from three chains were then combined and recorded as large samples of the full posterior joint distribution with a total of 30,000 sets of estimated parameter values. The above procedure was carried out independently for each model M1, M2, M3 and M4. The adjustment of these four models was compared according to the penalized-likelihood Deviance Information Criterion (DIC) [27,32]. The model with the best (lowest) value of DIC was chosen to explore the full distribution of posterior joint parameters obtained as output of rjags. For each parameter, we considered (i) the estimate as the median value of the posterior marginal distribution, denoted “point estimate”, and (ii) the 95% credibility interval (95% CI) from the quantiles of 2.5% and 97.5% of this distribution. These point estimates were used to evaluate the goodness of fit of the model by plotting adjusted data (using point estimates of model parameters) versus observed data. Finally, the full joint distribution was used to simulate the pH kinetics (estimate and 95% credibility band) under different conditions.

## 3. Results

### 3.1. Model Fit and Selection

For all four considered models and all parameters, convergence of the three MCMC chains were successfully checked by visual plotting and according to the chosen convergence criteria. The computed chains could then be considered as a large sample of posterior distribution of the parameters. The DIC criteria for comparing different models were computed and are gathered in Table 2.

In Table 2, on the one hand, the DIC was highest for models 1 and 2 (with two fixed parameters, or by fixing the stabilisation time θ), suggesting a low quality of adjustment of these models on experimental data compared to the others. On the other hand, for model 4 (without any fixed parameter), the DIC seemed to suggest a better fit, but probably was penalised by the high number of parameters to be estimated. Model 3 with n fixed showed the best (lowest) DIC, suggesting that it was the most suitable model among those tested for describing our data and estimating parameters. For this model, the goodness-of-fit plot showing comparison between adjusted and observed values, as well as the diagnosis of residuals, are shown in Figure 3. These edits showed a reasonably fitted trend between the observed and adjusted data, with no visual bias; the residuals did not show important outliers, except for one clearly underestimated point, probably due to technical errors in the experiments (Figure 3). Despite these errors, we decided to keep this point in the collected dataset (as seen in Figure 1 “Full dose of lactate—MAP2″) to ensure equilibrium in the experimental design (four sampling time points per condition).

### 3.2. Parameter Distribution

The point estimate of each model parameter, as well as its 95% credibility interval obtained with model 3, are gathered in Table 3.

#### 3.2.1. Stabilization Time and pH

As the output of model 3, the stabilisation time θ was estimated at 12.9 days (Table 3). Its posterior distribution suggested a 95% credible interval, varying from 12.1 to 13.7 days.

The pH reached at the stabilization phase can be deduced from Equation (1) to Equation (4), using the values of parameters (as estimated in Table 3). For example, for sausage samples formulated with 2% *w/w* (full dose) of lactate and packed under air, an average value of pH at the 15th day can be estimated as follows:$$\begin{array}{cc} & pH_{day15\;\left(stabilisation\;phase\right)}=-\beta_{air}*\theta +pH_{0}=-\exp\left(\lambda *Lactate^{n}+\delta_{Air}\right)*\theta +pH_{0}\\ & =-\exp\left(-0.095*2^{1}-2.430\right)*12.9+6.49=5.55 \end{array}$$

#### 3.2.2. Acidification Rate under Effects of Lactate and Atmospheres

The parameters of the model characterising the effects of lactate and atmospheres on acidification have been defined, for their respective prior, by non-informative uniform distribution. The experimental data enabled us to successfully estimate these parameters; their 95% credible intervals were much narrower than the prior, suggesting more targeted ranges of values. As shown in Table 3, the parameter λ characterising the effects of lactate on the acidification rate was estimated at −0.095 (95%CI: −0.119; −0.071). Its negative value suggested that increasing the initial lactate content in the formulation enabled one to decrease the acidification rate and, therefore, slow down the drop in pH. The additive effects of atmosphere were estimated, from lowest to highest, at −2.487 (95%CI: −2.587; −2.394) for MAP_1_:70%O_2_-30%CO_2_, −2.430 (95%CI: −2.528; −2.340) for air packaging and −2.339 (95%CI: −2.429; −2.254) for MAP_2_: 50%CO_2_-50%N_2_. These estimations suggested that, for a given sample of formulated sausages, packaging under MAP_2_: 50%CO_2_-50%N_2_ would accelerate the acidification phase in comparison with the air packaging and MAP_1_:70%O_2_-30%CO_2_. In order to better illustrate these effects, the variation in the acidification rate (β, covariate of the model) was calculated using the above point estimates of $\lambda ,\;\delta_{Air},\delta_{MAP1},\;\delta_{MAP2}$ and is shown in Figure 4.

The computed evolutions of β in Figure 4 showed that increasing the initial lactate content would slow down the acidification of turkey sausages, whatever the atmosphere used. The fastest acidification was observed in the case of ‘Zero lactate’ sausages, packed under MAP_2_: 50%CO_2_-50N_2_. Under these conditions, the acidification rate was estimated to be around 0.096 unit of pH per day; i.e., a drop of one pH unit in just about 10 days. In contrast, for sausages formulated with 2% *w/w* lactate and packed under MAP_1_: 70%O_2_-30%CO_2_, the rate was estimated at 0.068; i.e., a drop of one pH unit in 14.7 days. The slightly curved evolutions of the suggested effect of lactate, i.e., β, were not necessarily linear: for example, according to Figure 4, increasing lactate content from 0 to 1% *w/w* seemed to slow down acidification slightly more easily by comparing to the increase from 1% to 2%.

#### 3.2.3. Initial pH and Variability Sources

The mean initial value of pH across production batches $\mu_{pH}$ was estimated to be 6.49 on day 0, with a 95% credibility interval of 6.41 to 6.57 (Table 3). The estimated distribution of initial pH was significantly higher than the distribution of the prior, previously defined as a normal distribution, with a mean of 5.84. The standard deviation of the initial pH $\sigma_{pH_{0}}$ describing the variability across production batches was estimated to be 0.10. The parameter $\sigma_{pH}$ describing the random variations from the measurement at each time and each condition was estimated at 0.15. For these standard deviation parameters, the posterior distributions have shifted to the right of the prior distribution, revealing that the pH variabilities and uncertainties estimated from our data were greater than initially assumed in prior information (Figure 5).

### 3.3. Possible Use of the Model for Simulation

The full Bayesian procedure for the four models with all the estimated parameter distributions was prepared (R scripts/RData) and is available online at https://github.com/ndmluong/acidification (accessed on 11 March 2022). These files can be used as a first step to perform numerical simulations for the pH kinetics of sausages under different user-defined formulation-atmosphere conditions (pH curves and credible bands). Some simulation examples for two different initial formulations (0.5% and 1.5% *w/w*) are plotted in Figure 6. Such simulations can be useful for validation purposes on external experimental data, for example, by evaluating how external data could be overlaid by credibility bands simulated using our estimated parameters.

The provided scripts also allow the possibility to describe pH evolution and estimate acidification rates for other independent data, possibly collected on other food matrices, with other user-defined atmosphere factors and/or other formulations as covariates. Tutorials are available in the provided R script files to help the users get started with our modelling analysis procedure.

## 4. Discussion

In the present study, a comparative approach was performed, in order to select the most appropriate model (with alternatively fixed or estimated parameters) to describe a two-phase acidification curve, including a likely breaking point between two sampling times. The model selection, based on a penalized-likelihood criterion (DIC), must be a good compromise between improving adjustments and limiting the number of estimated parameters due to the small amount of data. Surprisingly, arbitrarily setting the stabilization time to one of the four sampling time points did not necessarily improve the selection criterion, compared to other models where the stabilization time was estimated. Indeed, the latter was estimated successfully between two sampling time points (the 8th and 15th day). One possible explanation could be that the chosen sampling time points (with 7-day intervals) were not suitable for studying pH kinetics (hence, for spoilage as well). Therefore, setting the stabilization time at one sampling time could have strongly biased the fit of the model. In our models, we assumed that the stabilization time was the same for all lactate and atmosphere conditions to avoid an important number of parameters, making their estimation more difficult. In further studies, the experimental designs (number and interval between sampling times) may need to be adapted for each condition. In such adapted designs, one could possibly improve the model by assuming different stabilization times, depending on the condition.

An advantage of Bayesian simulation techniques is the ability to build a hierarchical structure and easily communicate on it via an acyclic graph, as illustrated in Figure 2. Such hierarchical structures enabled us to overcome the problem of independence in the dataset, associated with the successive stages of meat production on an industrial scale. Besides, the Bayesian modelling approach precisely estimated the random differences across industrial production batches. Hierarchical Bayesian approaches have already been applied in food modelling, in particular, to separately estimate the biological variability and the uncertainty on the parameters [20,33,34,35]. Additionally, MCMC simulations for Bayesian modelling provides, as output, complete estimated distributions of the parameters that could be useful for further model adjustment on external experimental data collected by other researchers. R tools (scripts and tutorials) are provided for this purpose.

The Bayesian model illustrated herein confirmed the effects of lactate and atmosphere on pH kinetics, observed in our previous work [25], with a notable improvement in the description of lactate concentration as a quantitative variate, rather than a qualitative factor. Adding lactate into the formulation slowed down the acidification process, as expected, likely due to its role in inhibiting several microorganism-producing acids [36,37,38,39]. The gas composition used in MAP_2_: 50%CO_2_-50%N_2_ accelerated the acidification phase compared to the Air packaging and MAP_1_:70%O_2_-30%CO_2,_ regarding the acidification rates. The quick acidification in MAP_2_: 50%CO_2_-50%N_2_ is probably due to its high initial content of CO_2_ that favours acid-producing bacteria, such as lactic acid bacteria [1,9]. Another possible explanation could relate to the formation of carbonic acids (H_2_CO_3_), resulting from the important absorption of CO_2_ into the meat products [40,41,42]. Besides the main expected results above, our modelling approach brought three major improvements. First, the storage time and the initial lactate concentration were described as continuous-scale co-variates, but not as factors, thus, providing further possibilities to simulate the process. Second, the effects of lactate and atmosphere were quantified as the acidification rate associated with each condition, then representing much more comprehensive parameters, in an intuitive way. Finally, the Bayesian MCMC simulation techniques made it possible to evaluate credibility intervals around these effects. The modelling of the acidification rates could even have been improved further, by including the concentration of O_2_ and CO_2_ in the model.

The initial value of pH (day 0) estimated with our data (at 6.49) was higher than the pH observed by Lerasle et al. [26], giving a slight deviation for high pH values from the fitting line (Figure 1). This deviation could be explained by the composition of the sausages, such as the type of poultry meat (chicken or turkey) and their respective percentage. This difference in composition has been reported, for example, by Saucier et al. [43], who observed a higher pH in chicken meat (around 6.18) than in turkey (around 5.95). Nevertheless, despite such differences at day 0, our model was successfully adjusted on the overall dataset, since the Bayesian approaches take into account prior information and likelihood of data as well. The two-phase trend appeared to provide an accurate fit, despite outliers, possibly due to technical errors in the experiments. The linear decrease in pH in poultry meat products has been reported in the literature. For instance, Tovunac, Galić, Prpić and Jurić [36] suggested a linear decrease in pH in chicken frankfurters under different packaging types by fitting decreasing linear models to experimental data. The stabilization phase has been described in the literature due to the buffering effect in food products, such as meat [44]; Ellouze, et al. [45] modelled the buffering effect in the evolution of pH related to the total production of lactic acid by bacteria, using logistic-based sigmoidal models. In our study, the major difficulty remained in the low number of sampling time points, which could be inappropriate to fit this type of model, hence, our choice of two linear phases. Nevertheless, the two-phase (acidification-stabilization) model described well the pH evolution in fresh poultry sausages under the influence of lactate formulation and the type of MAP.

## 5. Conclusions

The kinetics of turkey sausage pH were monitored and successfully fitted using a Bayesian modelling approach, despite a limited number of experimental data points. The addition of potassium lactate combined with air or 70%O_2_-30%CO_2_ packaging slows down acidification of fresh turkey sausages and consequently limits potential sausage spoilage due to lactic acid bacteria. However, before applying such preservation conditions, we recommend also assessing the sensory qualities of the final product. The Bayesian modelling approach could be reused for other studies, especially in the context of empirical modelling framework of meat spoilage. For such a purpose, the Bayesian modelling workflow—and the script—have been provided. The prior distribution could be generated using existing data or previous studies, as done here, but also by seeking experts’ knowledge.

## Figures and Tables

**Figure 1 foods-11-01114-f001:**
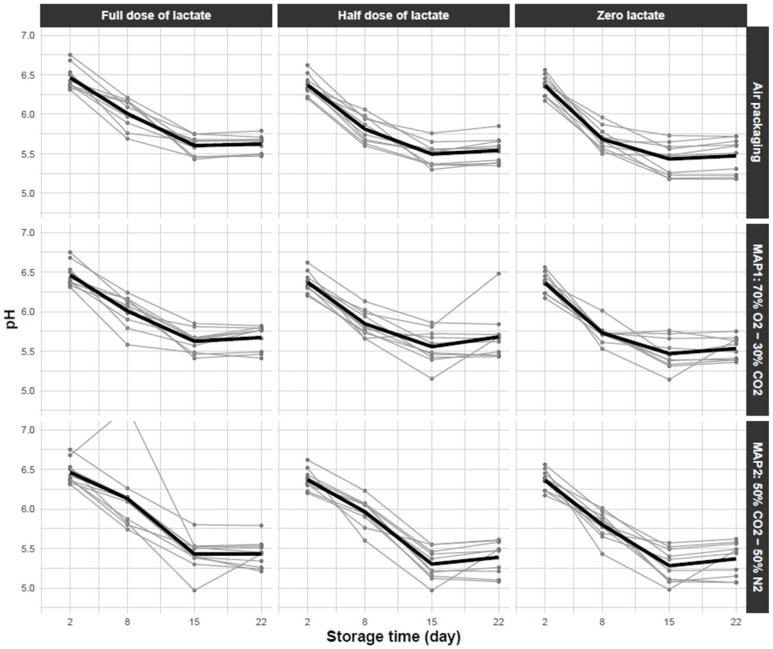
pH value of sausages as a function of storage time under different process conditions. For each condition, the points correspond to the average pH value obtained from three measurements on three different sausage samples (technical replicates), the different thin grey curves correspond to different production batches. The black thick curves correspond to the average pH across batches. Adapted from [25].

**Figure 2 foods-11-01114-f002:**
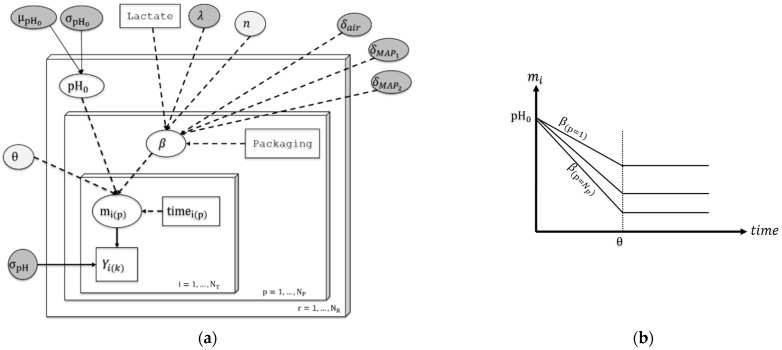
(**a**) Directed Acyclic Graph (DAG) of the model. The rectangles represent data or covariates and the ellipses represent model parameters. Directed arrows represents relationships between parameters, covariates and data: dashed arrows correspond to deterministic functions and solid arrows correspond to stochastic relationships. (**b**) Schematic representation of the deterministic part describing evolution of the average pH with different acidification rates under different process conditions.

**Figure 3 foods-11-01114-f003:**
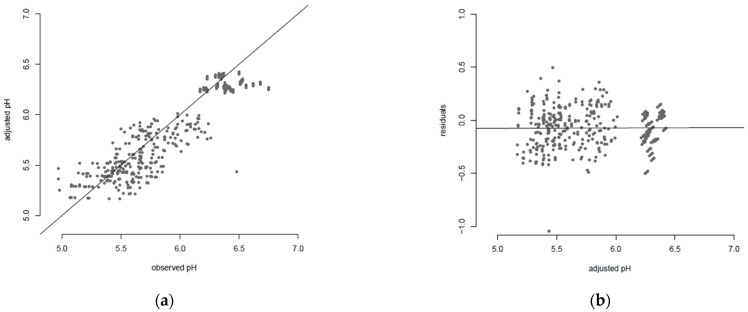
(**a**) Comparison of the observed experimental pH values with those adjusted by the model using point estimates of each parameter in model 3; (**b**) residuals versus adjusted pH values ($Residual=pH_{observed}-pH_{adjusted})$.

**Figure 4 foods-11-01114-f004:**
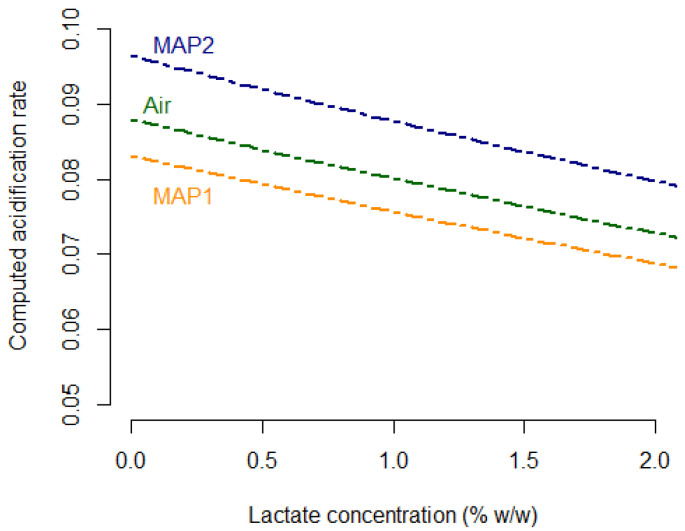
Acidification rate β computed using the point estimates of $\lambda ,\;\delta_{Air},\delta_{MAP1},\delta_{MAP2}$.

**Figure 5 foods-11-01114-f005:**
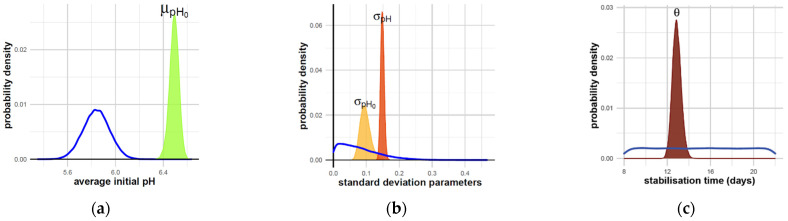
(**a**) Distribution of initial pH; (**b**) variability sources; (**c**) stabilization time. Prior distribution: blue curves for all parameters; posterior distributions: green curve ($\mu_{pH_{0}}$), light and dark orange curves ($\sigma_{pH_{0}}$ and $\sigma_{pH}$ ), red curve (θ).

**Figure 6 foods-11-01114-f006:**
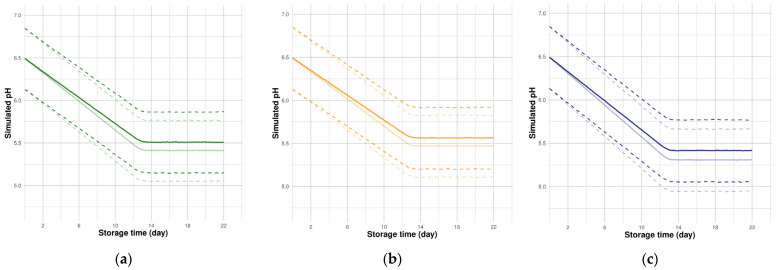
Simulation examples for pH kinetics of sausages formulated with two initial lactate contents (0.5% and 1.5% *w/w*) and packed under three atmospheres: (**a**) Air; (**b**) MAP_1_: 70%O_2_-30%CO_2_; (**c**) MAP_2_: 50%CO_2_-50%N_2_. Median curves (thick lines) and 95% credible bands (dotted lines). The transparency of the curves and the credible bands correspond to the initial lactate contents (the most transparent curves correspond to the 0.5% *w/w* formulation).

**Table 1 foods-11-01114-t001:** Parameters of the model: symbol, definition and prior distribution. *N* stands for the normal distribution, half-*N* stands for the half-normal distribution, *U* stands for the uniform distribution.

Symbol	Definition	Prior Distribution			
$\mu_{pH_{0}}$	Mean of the initial pH value across production batches	N(5.84, 0.11) *			
$\sigma_{pH_{0}}$	Standard deviation of the initial pH across production batches	half-N(0,0.1)			
$\sigma_{pH}$	Standard deviation of the pH value across measurement	half-N(0,0.1)			
$\delta_{Air}$	Additive effect of the “Air packaging” on acidification rate	U(−10;10)			
$\delta_{MAP1}$	Additive effect of the “MAP_1_:70%O_2_-30%CO_2_” on acidification rate	U(−10;10)			
$\delta_{MAP2}$	Additive effect of the “MAP_2_: 50%CO_2_-30%N_2_” on acidification rate	U(−10;10)			
λ	Slope parameter characterising the effect of lactate on acidification rate	U(−1;1)			
		Model 1	Model 2	Model 3	Model 4
n	Scale parameter characterising the effect of lactate on acidification rate	n=1 (fixed)	U(−3;3)	n=1 (fixed)	U(−3;3)
θ	Time point (in days) at which the pH reaches the stabilisation phase	θ=15 (fixed)	θ=15 (fixed)	U(8;22)	U(8;22)

* Lerasle et al., 2014 [26].

**Table 2 foods-11-01114-t002:** Deviance Information Criterion of the four considered models computed using the R package rjags.

Model	Total Number of Parameters to Be Estimated	DIC
Model 1	7 (*n* fixed, *θ* fixed)	−274.8
Model 2	8 (*θ* fixed)	−271.0
Model 3	8 (*n* fixed)	−296.5
Model 4	9	−294.1

**Table 3 foods-11-01114-t003:** Estimated parameters (model 3). The point estimate of each parameter corresponds to the median value of its posterior marginal distribution; the 95% credibility interval is defined by the 2.5% and 97.5% quantiles of the marginal distribution.

Parameter	Estimated (Point Estimate—95% Credible Interval)	
Stabilisation time		
θ (in days)	12.9	[12.1; 13.7]
Effect of lactate and atmosphere on acidification rate		
λ	−0.095	[−0.119; −0.071]
$\delta_{Air}$	−2.430	[−2.528; −2.340]
$\delta_{MAP1}$	−2.487	[−2.587; −2.394]
$\delta_{MAP2}$	−2.339	[−2.429; −2.254]
n	n=1 (fixed)	
Initial pH and variability sources		
$\mu_{pH_{0}}$	6.49	[6.41; 6.57]
$\sigma_{pH_{0}}$	0.10	[0.07; 0.13]
$\sigma_{pH}$	0.15	[0.14; 0.16]

## Data Availability

Publicly available datasets were analyzed in this study. This data can be found here: https://github.com/ndmluong/acidification, accessed on 14 March 2022.

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
