# Peer review of "A Bayesian Approach to Describe and Simulate the pH Evolution of Fresh Meat Products Depending on the Preservation Conditions"

_foods, 2022, doi:10.3390/foods11081114_

Round 1

Reviewer 1 Report

The purpose of this study is to use Bayesian model method to describe and simulate the change of pH value of fresh meat products with storage conditions. Taking several fresh poultry sausage with lactic acid formula as an example, the pH values of several industrial batches of fresh poultry sausage under three kinds of improved air (map) were measured. The results are interesting and valuable for predicting the preservation and quality of fresh meat. However, there are still some minor problems, as follows

1、    In line 26, it is said to add lactic acid to delay acidification, but in line 35, it is said that there is a conflict between the accelerated acidification of lactic acid produced by lactic acid bacteria fermentation and 81 adding potassium lactate to delay acidification. Please check carefully.

2、    Is it unreasonable that the sampling time in line 89 and the stable storage found in line 25 may be about 13 days? Please think and modify.

3、    Line 81 says that there are enough sausages in the same batch. I think the specific quantity should be specified to ensure representativeness.

4、    Line 76 said that the data used in this study came from a document monitoring several corruption reactions of fresh sausages with different lactic acid formulas. Can relevant documents be provided for researchers as supplement?

5、    FIG. 3 (a) shows the reasonable fitting trend between observed data and adjusted data, but some values deviate from the fitting line. Can such deviation be explained and discussed?

6、    The data values of each batch in Figure 1 were not analyzed for significance. Please check them carefully. Giving additional significance analysis is believed to be useful in the paper as well

7、    Can the pH drop rate in Figure 2 (b) be expressed as a numerical value when it reaches a stable level?

Reviewer 2 Report

I read the above article with interest. I am puzzled by a few things, a few are missing from the content and a few can be supplemented.
Here are the comments:

  1. I wonder where did the idea for such a system of experience come from? The addition of an acidifying substance (lactate) changes the sensory properties of the product. So it will be changed at the start. What about the acceptability of such a product? Because the model system should translate into the possibility of producing the product in industrial conditions (or at least on a semi-technical scale).
  2. Bayesian modeling consists in establishing the probability (degree of plausibility) of the conclusion h on the basis of the probability of premises and the degree to which the hypothesis makes the premises probable. I wonder if it really can be translated into a food matrix?
  3. Materials and methods: please provide me with the information related to the raw material composition of the sausages produced and the conditions of their thermal processing. The information that they are made of turkey meat is not enough. Each type of sausage deteriorates in a different way.
  4. I do not understand the data in Fig. 1: are these the results of the experiment or are they the findings of the experiment? If the results, they shouldn't be here.
  5. The summary/conlusions should be separated as a separate sub-item and it should be redrafted in terms of practical guidelines that may be applicable in practice.
  6. References: please review them carefully and the references with numbers: 4, 5, 9, 10, 16, 27, 34, 38, 39 and 45 should be replaced with newer ones. The 1982-1999 literature is far too old to be cited in 2022.
